# Lexicons of Key Terms in Scholarly Texts and Their Disciplinary Differences: From Quantum Semantics Construction to Relative-Entropy-Based Comparisons

**DOI:** 10.3390/e24081058

**Published:** 2022-07-31

**Authors:** Ismo Koponen, Ilona Södervik

**Affiliations:** 1Department of Physics, University of Helsinki, 00014 Helsinki, Finland; 2Centre for University Teaching and Learning (HYPE), University of Helsinki, 00014 Helsinki, Finland; ilona.sodervik@helsinki.fi

**Keywords:** quantum semantics, semantic networks, lexicons, text analysis

## Abstract

Complex networks are often used to analyze written text and reports by rendering texts in the form of a semantic network, forming a lexicon of words or key terms. Many existing methods to construct lexicons are based on counting word co-occurrences, having the advantage of simplicity and ease of applicability. Here, we use a quantum semantics approach to generalize such methods, allowing us to model the entanglement of terms and words. We show how quantum semantics can be applied to reveal disciplinary differences in the use of key terms by analyzing 12 scholarly texts that represent the different positions of various disciplinary schools (of conceptual change research) on the same topic (conceptual change). In addition, attention is paid to how closely the lexicons corresponding to different positions can be brought into agreement by suitable tuning of the entanglement factors. In comparing the lexicons, we invoke complex network-based analysis based on exponential matrix transformation and use information theoretic relative entropy (Jensen–Shannon divergence) as the operationalization of differences between lexicons. The results suggest that quantum semantics is a viable way to model the disciplinary differences of lexicons and how they can be tuned for a better agreement.

## 1. Introduction

Research fields always contain disciplinary groups where key scientific terms differ, and the same terms may be used differently in discussing and framing the key problems within the field. Such disciplinary fragmentation is particularly characteristic in the human and behavioral sciences [1,2]. Since scholarly texts are a major medium for disseminating the ideas and views of disciplinary groups, examination of how key terms are used in them provides important information on disciplinary differences. In this study, we suggest a simple method suitable for constructing, analyzing, and comparing networks of key terms (lexicons) in 12 scholarly texts related to learning sciences, to examine the differences and to find out how the lexicons could be tuned for better overlap.

The disciplinary structure of science and the formation of disciplinary schools, along with their characteristic ways of using scientific terms and forming various research programs, have long been topics of interest [3,4,5]. The use of networks as representations of the structure of science and its conceptual systems [4,5] was also recognized quite early, decades before the science of science [4] matured into the new research field of the cartography of sciences [6,7,8]. Contemporary science of science [9,10] has provided a variety of means to study the structure of sciences, their disciplinary and conceptual structures [6,7,8], and means for characterizing disciplinary differences [9,10,11,12,13,14]. Many such studies utilize methods based on text analysis and linguistic approaches developed within network science [15].

Focus on linguistic structures is also supported by studies within the philosophy and history of science. The language and linguistic structures involved in forming disciplinary groups, and their identity and ways of communicating scientific views and ideas, were central themes in Kuhn’s conception of science, which he developed after his views about scientific revolutions and paradigms [16]. In these later studies, Kuhn focused on the relational structures of scientific terms and concepts and how a networked, interlinked structure of concepts supports the formation of their meaning, referring to such networks as lexicons or lexical networks [16,17,18]. According to Kuhn, a lexicon is a network of terms, and terms are related through their connections in context. Here we adopt this view and take lexicon to correspond to the relatedness of words through their co-occurrence in texts, or rather, as will be seen later, how the co-occurrences are entangled in text. The basic assumption here is that disciplinary differences emerge from users of the lexicons having the option to use different interrelationships within the adopted lexicon. Such lexicons thus become basic structures characterizing the disciplinary groups and their identities.

Taking lexicons of scientific terms central for identifying disciplinary groups lends itself easily to the idea of using scholarly texts as sources to explore the similarities and differences in the use of terms between disciplinary groups within the same research field [15]. The semantic structure of texts and the semantic fields attached to their key terms are often approached using some version of co-word mapping or bag-of-words methods (see, e.g., reference [15] and references therein). Co-word mapping methods have a long history going back to seminal studies in the 80s and 90s [19,20,21,22,23], then still hampered by limited computing resources [15]. In the last two decades, co-word mapping methods have given room to the now widely applied topic modeling of texts, which has its origins in natural language processing and attempts to find topics on basis of analyzing bags of words from the text [15,24,25,26]. Consequently, word co-occurrence analysis and its different variants still find new and innovative applications in text and text-structure analysis [27,28,29,30].

Recently, text analysis methods that are more sensitive than co-word mapping to text structure have been introduced, with the aim of revealing syntactic and semantic structures that run deeper than co-word and semantic field analyses. The co-word mapping methods capture syntactic structures only up to word adjacency, but they can be generalized to take into account more complicated syntactic structures composed of subject–verb–object triads and networks constituted by such triads [31]. Other methods going beyond co-word analysis are looking in detail for example: concept diversity (cognitive content) through lexical diversity of text [14]; relation of key concepts and the architecture of the text structure [32,33]; searching semantic frames through finding communities of verbs and their arguments [34]; creating semantic networks by using concept-centered sub-networks [35]; and finding the larger-scale semantic structure of texts [36,37]. In all these cases, complex networks methods are used in construction of the semantic, lexical, or concept networks, and different network measures (closeness centrality, betweenness centrality, or some specially engineered measures for network topology) are utilized in analysis. Moreover, different types of computer software exist for different kinds of analysis methods—for example, for co-word mapping and topical analysis (for a reviews, see reference [15,37]) and analysis of semantic frames (for a review, see reference [34]). Here, however, the focus is not on the comparison of different methods or on providing a review of available methods and software, and therefore, more detailed discussion of such approaches and methods is beyond the scope of this study.

In this study, we propose a generalization of word co-occurrence counting to construct a network of key terms, a kind of lexical network, where links connecting key terms represent the probability of (generalized) co-occurrence. Following Kuhn, we refer such lexical networks as lexicons. The generalization we are proposing is meant to address the possibility of subjective differences in reading the meaning of a given sentence in a text. One potential way to overcome this shortcoming is a recently introduced method called quantum semantics, which provides a means to estimate the effects of subjective bias and entangled meanings [38,39,40,41].

Quantum semantics can be seen as a generalization of word co-occurrence counting methods or bag-of-words methods. It preserves the simplicity of co-word counting while allowing the inclusion of a factor to estimate the effect of subjective bias in reading the meaning of words in a bag of words [38,39,40,41]. From the perspective of quantum semantics, semantic fields are not fixed or unyielding structures that define relationships between words as extracted from texts, but rather, tentative templates for how producers of texts can be expected to compose sentences, and how readers of text can be expected to weigh the importance of co-occurrence of words in a sentence, i.e., read meaning into a sentence. In this study, we use quantum semantics in constructing the semantic networks (i.e., lexicons) that allow us to explore the extent of disciplinary differences in lexicons.

Seminal ideas of quantum semantics appeared already two decades ago and were applied to word association tasks and categorization tasks (see, e.g., reference [41] for a review), based on the insight that word meanings are entangled in the sense that the context where they appear with other words affects the ability to factorize meanings of words to elementary, independent elements (i.e., kinds of pure states). To describe such entangled states of word pairs, formalism from quantum logic, essentially in form of qubits, was adopted [41]. The recent idea to quantify the entanglement by using a quantum information theoretic measure called concurrence [38,39,40] provides substantial simplification to adopt quantum semantics as the basis to construct lexical networks. It should be noted that adopting quantum semantics to generalize word co-occurrence counting does not imply endorsing a view that words are quantum entities, only that generalized logic and probability behind the quantum theory are adopted. Here, we use such an approach to analyze scholarly texts to construct lexicons (lexical networks) of their key terms.

Quantum semantics makes it possible to study effective subjective bias between word pairs, and with that knowledge, to build a lexicon in the form of a (semantic) network. To explore the structure of lexicons, we invoke methods developed for analysis of complex networks. A network description is here based on a (weighted) adjacency matrix obtained through quantum semantics. The analysis is then carried out using an exponential matrix transformation that allows us to define a density matrix [42,43,44,45,46] for the lexicon (i.e., the probability of connectivity between links). The density matrix, on the other hand, when suitably scaled, can be interpreted as a correlation matrix between nodes in the network [47], i.e., now as correlations between co-occurrence of words when each pair of words is assumed to be entangled. Note that concurrence as a measure of entanglement between a pair of words or terms is the basis of obtaining the correlation matrix, but correlation matrix, as it is based on density matrix, takes all connections contained in the network into account.

The density matrix also allows a straightforward quantitative method for comparison of the lexicons, to be based directly on entropy differences between different lexicons. Here, the symmetric form of a relative entropy (here von Neumann entropy, because we are working with density matrix) in the form of Jensen–Shannon divergence [48,49,50] is adopted as the basis of comparison. Owing to its symmetric form and computational stability, Jensen–Shannon divergence is a convenient choice in comparing semantic networks and word distributions. Moreover, the square root of Jensen–Shannon divergence can be interpreted as a kind of (information) distance metric between networks [51,52,53].

To demonstrate the viability of this method based on quantum semantic networks in characterizing disciplinary differences of lexicons, we have chosen 12 scholarly texts concerning models of conceptual change (see reference [54] for a review). The texts deal with conceptual change models from different and even contradictory viewpoints. These texts are particularly suitable because the discrepancy of views they provide, to a shared research theme, is well known and discussed in the research literature. Therefore, it should be interesting for the reader to see if the method of analysis proposed here will reproduce the expected differences in level for lexicons, and in addition, whether or not the possibility to tune for better agreement of lexicons affects their similarity in essential and substantial ways.

It is shown that the method proposed here, although quite simple, indeed brings out clear differences between the texts, but also provides an understanding of how, through subjectively biasing a better overlap of their lexicons, mutual agreement could be possible. We believe that the methods introduced here may support the common traditional interpretative analyses, providing simple but sufficiently reliable quantitative measures to detect disciplinary differences in scholarly texts, and also providing a means to estimate a range of variation in differences.

## 2. Materials and Methods

The method to construct and compare lexicons consists of three methodologically different but closely interrelated steps:Quantum semantics is used to construct the lexicons corresponding a given text.The relevant connectivities of lexicons are found through network analysis.The comparison of lexicons is performed based on a quantum information theoretic version of Jensen–Shannon divergence.

We have limited discussions of methods to the minimum of what is needed to explain the steps in the analysis. The details of the derivation behind the results utilized here are well documented in the original references. The text corpora of scholarly texts used to test the method are 12 texts of conceptual change research. The texts were chosen to correspond to positions known to be seminal (three texts) texts displaying two opposing views (three texts for each case), and texts known to mediate between the opposing views (three texts).

### 2.1. Text Samples

The text samples chosen were 12 scholarly texts about models of conceptual change [55,56,57,58,59,60,61,62,63,64,65,66], published between the years 1982 and 2017. Short descriptions of the 12 texts selected for analysis are listed in Table 1 and identified with the acronyms provided. The selection was made on the basis of the widely held opinion that the research field of conceptual change has seen a long-standing dispute about the nature of conceptual change, in which some parties have tried to mediate between different views, but some other disciplinary schools have not found or even striven for a consensus. However, the basic terms and terminology and conceptions of the basic topic of interest are agreed on. In many ways, such a situation offers an interesting test-bed for a quantum semantics approach.

A well-known and widely discussed dispute is related to the question of whether conceptual knowledge should be seen as forming out of isolated and fragmented pieces (knowledge-in-pieces view according to some authors) or as broader and in some sense coherent theory-like structures (framework theories, according to some authors). In such disputes, referred to in what follows as “knowledge-as-elements” or “knowledge-as-theory” views (for reviews, see references [64,65]), the discussions are about how to frame the meanings of key concepts to be used in conceptual change. In this dispute, advocates of both views have held their positions and continued to discuss and criticize the opposing positions.

Of the twelve texts, texts V1–V3 belong to the camp of “knowledge-as-theory”, and texts D1–D3 are in the camp “knowledge-as-elements”. The texts OC, A1, and A2 are reviews, in which consolidation of different views is attempted. In particular, the texts A1 and A2 contain pleas for increased attempts to unify the terminology and strive for better mutual understanding in order to advance the research field. The texts Ca, P1, and P2 are seminal works that are often cited [54] but perhaps not so often directly used in developing researchers’ own positions. In this study, we do not discuss in greater detail the content of the texts or the disputes. Instead, we are interested in exploring whether the known opposing positions and attempts to mediate between positions are visible at the level of terminology and how terms co-occur in texts, i.e., at the basic level of semantic fields.

### 2.2. Generalizing Word Co-Occurrence: Concurrence

The first step in transforming the written texts into semantic networks, in the form of lexicons of terms, consist of splitting the sentences into clauses, in order of their appearance, and after that, recognizing the nouns, and within them, term-like words. The texts were first simplified by removing stop words and miscellaneous symbols. Next, nouns, verbs, adjectives, and attributes were picked out from sentences, preserving the sentence structure. Finally, resulting expressions were stemmed. In that, Mathematica 13 employing Porter’s algorithm for stemming [67] was used as software. While lacking advanced features of more advanced natural language processing (NLP), Mathematica produced a quite acceptable outcome and was selected here because it is supposedly easy enough to adopt, in comparison to more advanced NLP methods. This choice is a trade-off between accuracy and simplicity of analysis, because it is not likely that too complex and demanding methods will be adopted by researchers not working in field of NLP but who need to do similar kind of analysis in fields of education and behavioral sciences. After stemming, term-like words that appeared only once were discarded, and from the remaining set, about 70 were chosen for closer attention. Consequently, the selection process is quite simple and misses many finer points, but contains basic and robust elements commonly identified in analyzing speech and writing.

The second step consisted of finding the co-occurrence statistics for the 70 selected terms. To do so, we utilized quantum semantics approach [38,39,40], which generalizes the traditional word co-occurrence measures so that subjective factors can be taken into account at some level of idealization. In that generalization, the notions of entanglement of words and concurrence as a measure of the entanglement play key roles. Details behind the quantum semantic approach, as they are relevant here, are provided in Appendix A. The key result needed here is the concurrence *Q*, which quantifies the degree of entanglement of terms as they appear in a text. Symbols used in developing the methods are summarized in Table 2.

To obtain concurrence *Q* (degree of entanglement) between words A and B in a text, we count four different frequencies of co-occurrence: the frequency $n_{11}$, that A and B both occur in a given block of clauses at least once; $n_{00}$, that neither A or B occurs; $n_{10}$ that A occurs at least once but B does not occur; and $n_{01}$, that A does not occur but B occurs at least once. Such a situation can be described in the form of a classical 2 × 2 contingency table of association of two variables A and B. In quantum semantics, the association is described in terms of qubits, allowing for non-classical dependence due to entanglement (due to inherent inseparability and impossibility of factorization of co-occurrence). With these frequencies, the concurrence *Q* as a measure of entanglement is given as [38,39] (see also reference [68])
(1)$$Q=2\sqrt{n_{11}n_{00}+n_{10}n_{01}}\;\sqrt{1-2\;\Theta \frac{\sqrt{n_{11}n_{00}n_{10}n_{01}}}{\left(n_{11}n_{00}+n_{10}n_{01}\right)}}\;,\;\;\;0\leq Q\leq 1\;,$$
where −1≤Θ≤1 is the phase factor (see Appendix A for details and derivation) taken here as a free parameter to account for subjective bias [38,39,40]. For further reference, we define the prefactor $Q_{0}=2\sqrt{n_{11}n_{00}+n_{10}n_{01}}$ that corresponds to concurrence obtained with Θ=0. The last factor in Equation (Equation 1) given by $R=2\sqrt{n_{11}n_{00}n_{10}n_{01}}/\left(n_{11}n_{00}+n_{10}n_{01}\right)$ is the ratio of the geometric mean of frequencies $n_{11}n_{00}$ and $n_{10}n_{01}$ to their arithmetic mean. With these definitions, concurrence takes a form $Q=Q_{0}\sqrt{1-\Theta R}$. The entanglement is now possible only in cases R≠0.

For purposes of comparison, we note an analogy of concurrence to different factors measuring association between dichotomous variables in [69,70,71] (see also [38]). In classical methods, the deviation of associations corresponding to randomly shuffled and punctuated (random mixing) sentences leads to equivalence $n_{11}n_{00}=n_{10}n_{01}$ (representing the so-called odds ratio of value 1). Consequently, classical measures of contingency are often taken as either proportional to a factor $n_{10}n_{00}-n_{00}n_{11}$ (e.g., mean square contingency or Yule’s phi coefficient) or to $\sqrt{n_{01}n_{10}}-\sqrt{n_{00}n_{11}}$ (Yule’s Y coefficient). However, different other ways to measure the deviations from odds ratio are also used (see, e.g., [70,71] for a summary). There is no unambiguous way to define a contingency measure, but in all of them, deviations from an odd ratio of 1 (i.e., no correlations) is central. Consequently, in analogy with Equation (Equation 1), where phase factors *c* and frequencies *n* are related as $c=\sqrt{n}$, we define a modified contingency as
(2)$$C=2|\sqrt{n_{01}n_{10}}-\sqrt{n_{00}n_{11}}\;|\;,\;\;\;0\leq C\leq 1\;.$$
The modified contingency *C* as defined in Equation (Equation 2) fulfills now the normalization condition $\sum n_{ij}=1$ and corresponds to the minimum value of concurrence *Q* obtained with Θ=1. Therefore, it can be then compared with *Q* to illustrate the difference in quantum semantics-based quantification of co-occurrence to a minimal value of concurrence, which is closely related to contingency coefficients measuring traditional analysis of dichotomous variables. However, a more detailed quantitative match is not possible between concurrence *Q* and different classical contingency measures.

### 2.3. Constructing Lexicons from Concurrences

We are interested in the concurrence *Q* of all pairs of terms *i* and *j* in a set of 70 selected terms of interest. Each pair is then characterized $Q_{0}\left(i,j\right)=Q_{0}\left(j,i\right)$, along with factors R(i,j)=R(j,i). These factors are obtained for all pairs of terms in all texts. In what follows, we are interested in concurrence Q(i,j), tuned so that in two texts T and T′ the corresponding concurrencies Q(i,j) and $Q^{\prime }\left(i,j\right)$, respectively, become as close to each other as possible; i.e., we choose factors Θ(i,j) and $\Theta^{\prime }\left(i,j\right)$ so that difference is minimized. In what follows, *Q* always refers to such optimized concurrences as a best scenario for matching lexicons. The contingency *C*, on the other hand, corresponding to the minimal concurrence, provides the worst scenario as a benchmark. The pairwise values Q(i,j) and C(i,j) are used to form weighted adjacency matrices Q and C with elements ${\left[Q\right]}_{ij}=Q\left(i,j\right)$ and ${\left[C\right]}_{ij}=C\left(i,j\right)$, respectively, describing the connectivity of terms in the lexicon.

### 2.4. Characterizing Lexicons: Density Matrix

To explore the structure of lexicons, we introduce the weighted adjacency matrix **W** with elements ${\left[W\right]}_{ij}=W_{ij}$, where $W_{ij}$ is taken to be either Q(i,j) or C(i,j) for nodes *i* and *j* and $W_{ij}=0$ for unconnected nodes. In what follows, symmetric adjacency matrices are assumed. For a weighted adjacency matrix, it is recommended to use normalization [42,43,49,72] to obtain a normalized weighted adjacency matrix W, which for connected nodes has elements ${\left[W\right]}_{ij}=W_{ij}/\sqrt{d_{i}d_{j}}$, where $d_{i}=\sum_{j\neq i}W_{ij}$ is the weighted degree (strength) of node, while for unconnected nodes ${\left[W\right]}_{ij}=0$.

To assign a probability density to links in the network, we then introduce density matrix ρ characterizing the network [42,43,44,45,46]
(3)$$\rho =Z^{-1}\;\exp\left[\;\beta W\;\right]\;,$$
where Z=Trexp[βW] is a normalization factor. Due to normalization, Trρ=1, the matrix, has an analogous role to probability density; due to its constructions based on a real and symmetric matrix W is Hermitian; and finally, due to exponential matrix transformation, it is positive semidefinite matrix; therefore, it can be taken as a density matrix. Here, the weight factor β defines how links are weighted, with low values corresponding to situations where the network disintegrates into an unconnected set of nodes, and high values of β emphasize the role of strong links. The density matrix ρ can be also interpreted as path or walk counting, providing information on all paths or walks that connect the nodes in a network [42,43,44,45,46]. The counting of walks is closely related to the question of how nodes can be reached in given networks, through connecting links. Therefore, we can use density matrix in Equation (Equation 3) to establish a connection between a correlation matrix, correlating “positions” of nodes in the network, “position” meaning how through different paths nodes can be reached. In what follows, a short summary is provided, and details not essential for the rest of the study are relocated to Appendix B.

The density matrix ρ provides a way to formulate a correlation matrix describing how “positions” of nodes in the network are correlated [47]. Following Estrada’s derivation (see Appendix B and reference [47]), one can show that it is possible to define a correlation matrix Γ in the form
(4)$$\Gamma =\gamma^{-\frac{1}{2}}\rho \;\gamma^{-\frac{1}{2}}\;,$$
where γ=Diag[Γ] is a diagonal matrix. The elements of ${\left[\Gamma \right]}_{ij}$ of the correlation matrix can be shown to be directly related to the covariance of values of nodes *i* and *j* in the network (see Appendix B and reference [47]), providing a way to find the key nodes on the basis of correlations. Towards this end, we define a correlation centrality as an off-diagonal sum:(5)$$\Gamma_{i}=\sum_{j\neq i}\;{\left[\Gamma \right]}_{ij}\;,$$
which closely resembles communicability centrality [42,43]. The correlation centrality is used here as a basis to define key terms and their rankings.

### 2.5. Comparing Lexicons: Divergence and Similarity

Next, we turn to the task of comparing lexicons described by different density matrices ρ and σ. An obvious approach is to use information theoretic relative entropy as a measure of difference (see, e.g., [48,49,50,51,52,53]). Relative entropy quantifies the amount of information that is needed to make inferences about a given network using information contained in the network as a basis of comparison. Here, we use the Jensen–Shannon divergence (relative entropy), which is widely used in the comparison of semantic networks [51,52,53]. Most often, in applications to semantic meaning of words, Jensen–Shannon divergence is applied in context of ordinary probability densities [51,52,53], but can be generalized for density matrices, as far as different entropy functions can be generalized for density matrices (which is usually accepted as in classical case of generalizing Gibbs–Shannon entropy as von Neumann entropy in quantum mechanics) [73,74,75,76].

Jensen–Shannon divergence (JSD) is based on von Neumann–Shannon information theoretic entropy H(ρ)=−Trρlogρ and is defined as a symmetric relative entropy between density matrices σ and ρ in the form [73,74,75,76] (see also [48,49,50])
(6)$$J\left(\rho ||\sigma \right)=H\left(\frac{\rho +\sigma }{2}\right)-\frac{1}{2}\left(\;H\left(\rho \right)+H\left(\sigma \right)\;\right)\;,\;\;0\leq J\leq 1\;.$$

In characterizing systems that can be represented in terms of density matrices, Jensen–Shannon divergence has several convenient properties: It is positive, zero if and only if ρ=σ, symmetric, and always well-defined. In addition, the square root $\sqrt{J(\rho ||\sigma )}$ of the Jensen–Shannon divergence can be also interpreted as an information-based metrics for the distance between σ and ρ [75,76]. Nevertheless, in practice, computation of divergence requires some caution, because it involves computing a matrix logarithm. To overcome issues related to the stability of computation, we have used so-called hypoentropy [77]: $H_{\lambda }\left(\rho \right)\propto -\left[\mathrm{Tr}\;\left(I+\lambda \rho \right)\log\left(I+\lambda \rho \right)\right]/\lambda$, which has the limiting values $H_{\lambda }\to H$, when λ→∞, thus providing von Neumann entropy [77].

The similarity of a network described by ρ to a network described by σ can now be defined through Jensen–Shannon divergence (JSD) as
(7)$$S_{\mathrm{JSD}}\left(\rho ||\sigma \right)=1-\sqrt{J(\rho ||\sigma )}\;,\;\;0\leq S_{\mathrm{JSD}}\leq 1\;,$$
where the square root of JSD is used because it can be interpreted as a metric, as noted before. This kind of a similarity measure is an example of information-theoretic similarity measures (compare, e.g., [78].

For comparisons, we use a different type of similarity, based on correlation centrality values of nodes. By forming for lexicons L and L′ centrality vectors $\bar{\Gamma }$ and ${\bar{\Gamma }}^{\prime }$ consisting of centralities $\Gamma_{i}$ and $\Gamma_{i}^{\prime }$ of nodes, respectively, we can define the so-called cosine similarity $S_{\mathrm{COS}}$ as a dot-product [79,80]
(8)$$S_{\mathrm{COS}}=\left(\;\bar{\Gamma }\cdot {\bar{\Gamma }}^{\prime }\;\right)\;/\;\left(\;|\bar{\Gamma }\left|\right|{\bar{\Gamma }}^{\prime }|\;\right)\;,\;\;0\leq S_{\mathrm{COS}}\leq 1\;.$$
The cosine similarity $S_{\mathrm{COS}}$ amounts to comparing the importance of key terms in lexicons. When it attains a value 1, networks are completely similar, as characterized by vectors of values of correlation centralities of nodes, whereas for completely different networks, cosine similarity has a value 0. As the correlation centralities of nodes are obtained as off-diagonal sums of the correlation matrix, related to density matrix, we can expect that similarity $S_{\mathrm{JSD}}$ based on Jensen–Shannon divergence and cosine similarity $S_{\mathrm{COS}}$ provide results that are in agreement, although they may differ slightly; they provide complementary information on the origin of the similarity.

## 3. Results

The method to construct, analyze, and compare lexicons was demonstrated by analyzing 12 scholarly texts related to conceptual change (see Table 1). First, we discuss results on the basic features and statistics of lexicons. Second, results in identifying key nodes are introduced, and third, on the basis of these results, the similarity of lexicons is discussed. Fourth and finally, key terms responsible for the similarity of lexicons are presented.

### 3.1. Word Frequency Statistics

The text analysis and stemming was carried out by using Mathematica 13, as explained in previous section. The texts were of different lengths, and they contained different numbers of words and sentences. The numbers of sentences and words after stemming in each text are provided in Table 3. Although the absolute numbers of sentences and words varied quite a lot, the ratio of words to sentences was in range from 8 to 16; the mean value and standard deviation were 12±2. In what follows, only relative frequencies of co-occurrence are of interest; absolute numbers are of no further relevance.

Of the list of stemmed expression, the most relevant 150 words and terms were selected for further scrutiny. Selection was done on basis what appeared to be of relevance for the topic (conceptual change, learning, and instruction) in question. For example, “earth” and “hollow”, though occurring in many documents, were dropped, since they are related to one specific example often discussed, but not of interest to discussing the general nature of conceptual change. From the resulting list, we then selected 90 terms having the highest frequencies of appearance for closer scrutiny, to be analyzed through their co-occurrence statistics.

Eventually, on basis of co-occurrence, only about 50 out of the 90 terms appeared to be are of interest. These terms are also listed in Figures 7 and 8, where we can see that most of them, but not all, are obviously relevant for the topic of conceptual change. The frequency distribution of the 50 most frequently occurring terms in all 12 texts is shown in Figure 1 (panel a, left). The distributions are shown as functions of rank of occurrence; the most common term has the highest rank of 1, and the larger the rank number, the lower the frequency. The term “concept” is omitted, because its appearance is dominant, being about two times more frequent than the next most abundant term, “student.” The distribution of terms, being flat, shows how no specific term appears preferentially in the 12 texts chosen as examples. This is also seen in individual distributions (Figure 1b, right, in the same order of rankings as in the large figure a at left) shown separately and denoted by acronyms (see Table 1). As can be seen, a few terms stand out in each individual text as the most abundant, but they are different ones in different texts. This reveals the diversity of texts even just on the basis of word frequency distributions.

### 3.2. Lexicons: Concurrence and Construction

To get a more in-depth picture, we next turn to the (modified) contingency *C* and concurrence *Q*, and construct the lexicons in a form of a network. The concurrence *Q* is of special interest, because optimizing it allowed us to tune the lexicons to overlap better, and thus led to more similar semantic fields. As was explained previously (Section 2.3), in tuning lexicons for the best overlap, we found for each link a phase factor which brings the concurrences as close to the their common average value as possible (constrained only by a factor *R*). For each pair of lexicons, and for each pair of links in each lexicon, these phase factors are different.

Figure 2 shows three examples of how optimized concurrences *Q* and corresponding contingencies *C* are related for lexicons A and B. The examples in Figure 2a–c (in upper row) show that contingencies *C* (gray symbols) for connections between word/term pairs in different lexicons A and B are not related to a significant degree. The situation changes when one turns to optimized concurrence *Q* (black symbols), when for many (but not for all) term pairs it is possible to find a phase factor Θ that brings the concurrences of term pairs in lexicons A and B close to each other; in many cases, values of *Q* in lexicons A and B are aligned, forming a straight line, as seen in Figure 2a–c. This demonstrates that optimization significantly improves the overlap of lexicons, and consequently, the necessary conditions for the formation of shared meanings. For example, in cases (Figure 2a) of A = A1 tuned to B = V2 and D1 (some of the best matching cases), we can see a significant increase in how concurrences are aligned. In the case (Figure 2c) of A = V3 tuned to B = D3 (one of the worst-matching cases), the tuning does not lead to equally high optimal values of *Q*, but still much higher values than obtained for contingency *C*, indicating a significant increase in the overlap of lexicons even in that case. The results in Figure 2 demonstrate the effect of optimization and differences between optimized concurrences and contingencies of term pairs. We will later (Figures 7 and 8, having first discussed the similarity of lexicons) return to the question of which terms are those ones yielding optimization, and at the same time, having significant roles in increasing the similarity of lexicons.

A complementary picture of the effects of tuning is provided in Figure 2d–f, where the same cases as in Figure 2a–c are shown differently, plotting the values of concurrence *Q* against the values of contingency *C* in each individual lexicon. In the results in Figure 2d–f, we can observe that while some terms have same values of *Q* and *C*, most of the terms have widely different values, and in general, values of *Q* are higher than those of *C* (the cases A = V3 and B = D3 being particularly clear). This is, of course, as expected, provided that *Q* is tunable. In summary, from the results in Figure 2, we can conclude that many terms yield optimization (points above the straight line in Figure 2a–c), and these highly tunable term pairs play an important role in improving the overlap of lexicons and providing shared meanings.

To provide an idea of the diversity of lexicons, we show in Figure 3 all 12 cases as projections on an underlying spring-embedding of agglomerated networks of all 12 cases. In this visual rendering of the lexicons, the size of a node (term) is proportional to its correlation centrality as defined in Equation (Equation 5). The individual lexicons, as shown in Figure 3, already show through visual inspection that they differ in size but show little systematicity, thereby defying simple classifications or categorizations. Visual representations, as shown in Figure 3, are suggestive, but quite unreliable for making inferences about differences. Therefore, we turn to more controllable ways to estimate the similarity of lexicons.

### 3.3. Divergence and Similarity of Lexicons

Networks appear differently for different values of weight parameter β, and to decide when a stable region is reached and networks can be compared, we need to monitor the behavior of divergence when the value of β is increased. Examples of the behavior of Jensen–Shannon divergence *J* (based on concurrence *Q*) with increasing β are shown in Figure 4a–c (upper row) for lexicons V1, OC, and A1. In each case, the 11 different sets shown correspond to the results when a given lexicon is optimized for the best overlap with all the other 11 lexicons (denoted in the legends in upper row). In all cases, we can see that divergence grows rapidly in region 2<β<5, reaches a maximum at 5<β<10, and eventually reaches constant values for β>30. This behavior results when increasing β, links with different weights contribute to the divergence. In regions with of low values of β, all links are weak, and since networks contain always same nodes, they are essentially disintegrated but similar. Gradually, with increasing values of β, many links begin to gain importance with increased weighting, and many of them begin to contribute to the similarity (or rather, dissimilarity) of networks; divergence reaches a maximum. With increasing weighting, however, only the most important links with the highest values of concurrence continue to contribute to the similarity and divergence decreases. Such a decrease in divergence indicates that lexicons possess, after all, quite a few term pairs connected by high concurrence links. Therefore, with the most important links dominating the behavior of the divergence at the highest values of β, divergences do not change anymore. In this region, one can reliably identify the key nodes (terms) and their rankings.

The divergences *J* for optimized lexicons show large variation. Lexicons OC (Figure 4b) and A1 (Figure 4c) have much lower divergences in comparison to V1 (Figure 4a). Such differences cannot be observed in the case of non-optimized lexicons corresponding to modified contingency *C* and shown in Figure 4d–f (lower row). The comparison of results in Figure 4a–c to those in Figure 4d–f (note the difference in scales) for *Q*- and *C*-based divergences demonstrates the important effect of optimization on matching the lexicons. However, the relative ordering of divergences is somewhat similar in both cases; optimization increases match, but does not dramatically change how relative differences behave.

Jensen–Shannon- divergence *J* (JSD) provides a basis to define the similarity $S_{\mathrm{JSD}}$ of lexicons, as introduced in Equation (Equation 7). The similarity $S_{\mathrm{JSD}}$ takes into account all links in the lexicons, and thus, is a global similarity measure. The cosine similarity $S_{\mathrm{COS}}$ in Equation (Equation 8), on the other hand, is based on the correlation centrality of nodes, i.e., off-diagonal sums of the correlation matrix. Figure 5 shows the similarities $S_{\mathrm{JSD}}$ and $S_{\mathrm{COS}}$ (upper and lower panels, respectively) for C- and Q-based lexicons (left and middle panels, respectively) as deviations from the median similarity (values shown above legends). The absolute change ΔS in similarity when C-based lexicons are optimized to Q-based lexicons is shown in Figure 5 in the right-hand panels.

Results in Figure 5 for $S_{\mathrm{JSD}}$ show that in all cases the increase in similarity due to optimization is significant, although relative patterns as related to median values are somewhat similar, as already seen in three examples in Figure 4. However, some interesting changes in relative patterns are notable. The clusters V1–V3; D1–D3; and OC, A1, and A3 are also present in contingency-based similarity patterns (upper row, left), but D1 is not strongly featured in the cluster D1–D3. It is also noteworthy that A1 in Q-based similarity is strongly similar to all other lexicons, but not to the same degree as in C-based similarity. However, in all cases all similarities increase significantly when lexicons are changed from C-based (median similarity 0.61) lexicons to Q-based ones (median similarity 0.94) for better matching.

It is interesting to note, as Figure 4 and Figure 5 reveal, that texts V1, V2, and V3 are by the same lead authors, and thus supposedly belonging to the same thematic area and topics, so they are similar to each other. However, e.g., V1 is most similar to V2, yet it is more similar to OC and A1 than to V3. This indicates that V1 and V3 differ somewhat in their vocabularies, because apparently the authors wrote about their topic with differing emphasis, but nevertheless, review articles OC and A1 with the purpose of covering all aspects as thoroughly as possible, attained even larger similarities. In concordance with that notion, lexicon OC corresponding to a review article is most similar to V1 and V2, and next to them to A1, which is also the lexicon of a review article. Lexicon A1, on the other hand, is most similar to A2, OC, and V2, in that order. Contrary to these more or less expected findings, Figure 5 shows that sets D1, D2, and D3 of lexicons are less similar to each other than they are to lexicon A1; the authors of D1, D2, and D3 have to some degree different vocabularies in their texts, but the review article manages to cover all of them. In addition, V1, V2, and V3 all have quite low similarity to D1, D2, and D3, which was expected, because the main dividing line of thought is known to be between these disciplinary sub-groups. There are other more minor details of some interest, but for this study attempting to demonstrate the viability of the proposed method, it is not necessary to discuss those differences or their origins in more detail.

Scatter plots of similarities $S_{\mathrm{JSD}}$ and $S_{\mathrm{COS}}$ for all 12 lexicons are shown in Figure 6, based either on contingency *C* or concurrence *Q* (left and in the middle, respectively, in Figure 6). In scatter plots, Kendall’s tau (non-parametric) correlation coefficients τ for the data points are also provided. The results in Figure 6a,b (upper row) are shown for β=0.3 and for a stabilized state with β=50 (Figure 6d,e in lower row). It is seen that at low values of β, the correlation between *Q*- and *C*-based cos-similarities $S_{\mathrm{COS}}$ is larger than for JSD similarities $S_{\mathrm{JSD}}$. For comparison, straight lines indicate the reference lines of equal values, to show an increase in similarity owing to the optimization (i.e., all data points are well above the reference lines). As cos-similarity puts weight on key terms, the behavior shown in Figure 6 indicates that key terms contribute significantly to similarity. Lower values of correlation for JSD-based similarities in comparison to cos-similarity indicate larger variety in weak links corresponding to auxiliary terms. This is a conclusion that can also be inferred from the examples shown in Figure 2. The results in Figure 6c,f on the left display how cos- and JSD-similarities are correlated for *C*- and *Q*-based lexicons at low and high values of β. We can see that in the stabilized region, *Q*- and *C*-based similarities align, while in initial, low values of β, the scatter of values is much larger. The main message of this figure is that the best correlation of similarities is achieved between *C*- and *Q*-lexicons in stabilized region β≫1, meaning that in this stage, the key terms of optimized lexicons dominate the similarity. In this region, cos-similarity and JSD-similarity provide essentially the same results.

From the results in Figure 5 and Figure 6, we can conclude that some lexicons of some texts are highly overlapping. For example, lexicons V1 and V2 are quite similar to each other, as are also lexicons P1 and P2 and A1 and A2. The texts that correspond to these lexicons had the same authors. In addition, lexicon A1 is highly similar to nearly all other lexicons, and OC also overlaps with many of the lexicons. These lexicons are from review-like texts, and A1 in particular attempts to cover many different views on conceptual change. In optimization, the similarity gain is also largest for the lexicons A1 and OC.

Finally, it is of interest to compare the similarity of lexicons based on co-occurrence counts to the similarity obtained on the basis of frequency distributions shown in Figure 1. Such a comparison can be carried out using JSD-similarity, which can be calculated from Equations (6) and (7) for ordinary probability distributions by replacing density matrices with probability density distributions and traces with sums. The resulting pairwise similarities of lexicons are similar to results shown in Figure 5 and Figure 6, and again the same high similarity clusters, as discussed above, can be discerned. For comparison, the Kendall–tau correlations of frequency-based lexicon JSD-similarities to concurrence and contingency-based similarities are now 0.37 and 0.35, respectively. The differences are not large, indicating that few frequently appearing terms are also the terms that connect to other frequently occurring terms. Next, we turn to identifying the key terms responsible for the similarity and how they are shared in different lexicons.

### 3.4. Key Terms and Shared Key Terms

A rough idea of the rankings of key terms is provided in Table 4, where the top-most key terms are listed in cumulative order, for *C*- and *Q*-based lexicons and for low and high values of β. The key terms are listed in four cohorts. Cohort I contains all terms that are found among the top 10 terms in at least eight lexicons, cohort II contains terms found among the top 20 but not among cohort I, and similarly cohorts III and IV contain terms that appear in the top 40 and 60 but not in any of the lower cohorts. From the results in Table 4, we can see that in *C*-based lexicons, cohort I contains only a single shared term, in contrast to *Q*-based lexicons, where several terms are shared.

This demonstrates the significance of optimization. In cohorts II–IV we can see many differences when *C*- and *Q*-cohorts are compared. Although the collections of terms are nearly the same, their positions within cohorts changed. Terms that appear in higher Q-cohorts in comparison to their positions in C-cohorts are the terms that need to increase their value of concurrence with other terms due to optimization (e.g., terms “conceptual”, “change”, “knowledge”, and “theory”) to become better shared, and very few terms (e.g., “nature”) were moved lower cohorts due to optimization. The rankings of top terms in *C*- and *Q*-based lexicons are thus different, although the differences are not dramatic. It is noteworthy that in both *C*- and *Q*-lexicons, the effect of β on the rankings of the upper-most terms was not very significant. This is in agreement with the result that cos-similarity (which weights key terms) did not change dramatically with increasing value of β (see Figure 5 and Figure 6).

Taking the *Q*-based correlation centrality as the basis of the rankings, the occurrences of the top-ranking terms in different lexicons are shown in Figure 7. The first panel, Figure 7a, shows how frequently a given term (listed on the right) occurs among the top 10 terms. If, in a given lexicon, the term is found among the top 10 ranking terms in all 11 optimized lexicons, the symbol is dark purple; the fewer the occurrences, the lighter the blue symbol. The occurrences among the 20 and 40 top ranking terms are shown in panels Figure 7b,c, respectively; and finally, in panel Figure 7d, for the 60 top ranking terms. Only cases where a given term is found in eight lexicons are shown.

In all texts, “concept”, “conceptual change”, “change(e)”, “student”, and “learn(ing)” are among the top 10 terms; and “knowledge” and “theori(es)” are among the top 10 in most lexicons. Other terms occur more sporadically, and one has large variability between lexicons in the distribution of top terms. When attention is based on occurrence among the 20 top terms, new frequently-occurring terms emerge. Among them are “science” and “research(ing)”. Furthermore, “model” can be found among the top 20 terms in many lexicons. In the set of 40 top ranking terms, new frequently-occurring terms are “structure”, “coherence”, “belief”, and “misconception”. This shows how different common themes emerge gradually as expanding shared sets of terms; the lexicons are overlapping. Finally, in the set of the 60 highest ranking terms, two new blocks of interest emerge. One block corresponds to framework theories with terms “frameworktheori(es)” and “coher(ence)”; the other block consists of knowledge-as-elements view terms “knowledge(in)piec(es)” and “p-prim”. In addition, the terms “process”, “exper(ience)”, and “explanation” are featured among the top 60 terms.

The breakdown of terms in cohorts of 10, 20, 40, and 60 top-ranking cohorts already gives us an idea how terms and their semantic fields are shared by different optimized lexicons, leading to the overlap and similarity of the lexicons. The top-ranking terms listed in Figure 7 are also the terms denoted by large nodes in Figure 3. Together, the information provided by these representations establishes a qualitative idea of which lexicons are similar and on what basis of shared terms.

The breakdown of key terms in top-cohorts as based on concurrence *Q* can be compared with similar breakdown when *C*-based correlation centrality is used as the basis of ranking. The breakdown in top cohorts in that case is shown in Figure 8. Comparing the results in Figure 7 to results in Figure 8, we see that with contingency *C*, key terms are not shared as frequently as for concurrence *Q*; many lexicons that overlap by sharing key terms with *Q* drop out for *C*. Comparison in Figure 7 and Figure 8 finally answers the question of which terms are the significant, tunable terms that are responsible for the increased similarity of lexicons; they are the terms appearing in Figure 7 but not in Figure 8. In cases of cohorts of top 10 and 20, we can locate several such important, tunable terms. On the other hand, the terms that appear in Figure 7 and Figure 8 both, i.e., in practice nearly all terms in Figure 8, are not tunable, or their roles are not changed by tuning. Therefore, in seeking better overlap of lexicons, terms appearing in cohorts of top 10 and 20 terms in Figure 7 but not in Figure 8 are the most important ones.

## 4. Conclusions

In this study, we have introduced an approach to constructing and comparing lexicons of terms in scholarly texts. The purpose of the method is to provide a relatively simple means to compare disciplinary differences as they appear in texts, and moreover, to study the extent to which lexicons can be optimized for the best possible overlap. The approach suggested here consists of three steps, each of which warrants some brief discussion.

First, we have adopted and utilized the recently suggested quantum semantics [38,39,40] to construct and optimize lexicons. Quantum semantics allows us to describe how subjective biases affect connections between words, and thus, give the possibility of different readings of the meanings of words. According to quantum semantics, the meanings of words and terms are entangled so that different meanings can not be entirely separated [38,39,40]. The “quantumness” in this context does not refer to any physical-like quantization, but rather, to a specific kind of a logic behind quantum physics, which allows the superposition and entanglement of states (here, co-occurrence of words). Such a logical structure is provided by quantum logic, and therefore, the approach borrowing it is referred to as quantum semantics. A major shortcoming of quantum semantics as applied to a written text is the ambiguity that remains in inferring the phase factors (see Appendix A) that determine the degree of entanglement [38,39,40]. Due to that limitation, compound phase Θ remains a fitting parameter. Although in principle the individual phase factors can be obtained (see reference [39] for a discussion), it is impossible with the sample and data in focus here. Therefore, we have only discussed results corresponding to extreme cases with classical contingency *C* (no optimization) and complete optimization of concurrence *Q*. This shows the possible bounds of variation of achieving overlap in lexicons, and thus, possibilities for improved agreement of meaning as far as it is reflected in similar co-occurrence of words in expressions. Despite many limitations, quantum semantics thus provides an alternative to approaches in which semantic fields are constrained to have fixed meanings, without the possibility of subjective bias. Interestingly, recent advances in quantum semantics [81] may open a way to discuss the role of subjective experiences (e.g., including affective factors) in determining the phase factors, and thus, tuning of lexicons. This possibility aligns well with approaches where sentiment analysis is included as part of analysis of cognitive networks (see, e.g., reference [31]. However, further discussions of such extensions is beyond the scope of present study.

Second, on the basis of quantum semantics, lexicons of terms were constructed in the form of networks and analyzed by utilizing path (walk) counting in terms of exponential matrix transformation, which is widely used as the basis of network analysis [42,43,44,45]. Based on exponential matrix transformation, a density matrix and correlation matrix [47] were introduced to characterize the lexicons. The correlation matrix allowed us to define the correlation centrality of nodes, used here to identify the important nodes in the lexicon.

Third, and finally, by using density matrices, comparisons of lexicons could be based on information theoretic relative entropy (Jensen–Shannon divergence, JSD) [48,50,51,52,53]. Apart from the use of a correlation matrix, as suggested in reference [47], the other steps were standard ones from other similarly focused studies, thereby requiring no further discussion.

The method suggested here was tested by applying it in the analysis of 12 scholarly texts about research in conceptual change. The results show that the method is able to bring out the disciplinary differences, in agreement with the expectations and known differences in the positions taken by different authors. That finding in itself was of course not unexpected, and many features revealed by the semantic analysis were probably easy to anticipate for those familiar with the content of the papers (see, e.g., reference [54] and references therein). It was also found that although concurrence, contingency, and frequency-based similarities produce different results, the differences are not dramatic; in all cases nearly the same clusters of texts turned out to be the most similar to each other. This indicates that frequently occurring terms are also the terms responsible for the strongest links between term pairs. Such a dependence is expected on the basis of tight focus of texts, and furthermore, a certain idiosyncratic way of authors discussing the topics.

Nevertheless, the results suggest that overlap of lexicons can be improved by optimization (i.e., tuning for best agreement). If one takes the lexicons central for communicating ideas across different disciplinary groups, as in Kuhn’s view [16,17,18], the ability to improve overlap can be taken as a proxy to enhance and improve communication of ideas and find a consensus of views.

In summary, we have demonstrated that a relatively simple (but many stepped) method of analysis which is a generalization of word co-occurrence counting is a viable way to construct and compare lexicons. It brings out the disciplinary differences in scholarly texts, and thus can be used to study differences in how various disciplinary schools frame their key concepts and use them in disseminating their ideas. We believe that having a sufficiently simple method for such analyses will help to lower the threshold to adopt network-based methods in fields of research (e.g., educational and behavioral science) where such methods are not yet well known, but where they might provide significant support for more traditional interpretative research approaches.

## Figures and Tables

**Figure 1 entropy-24-01058-f001:**
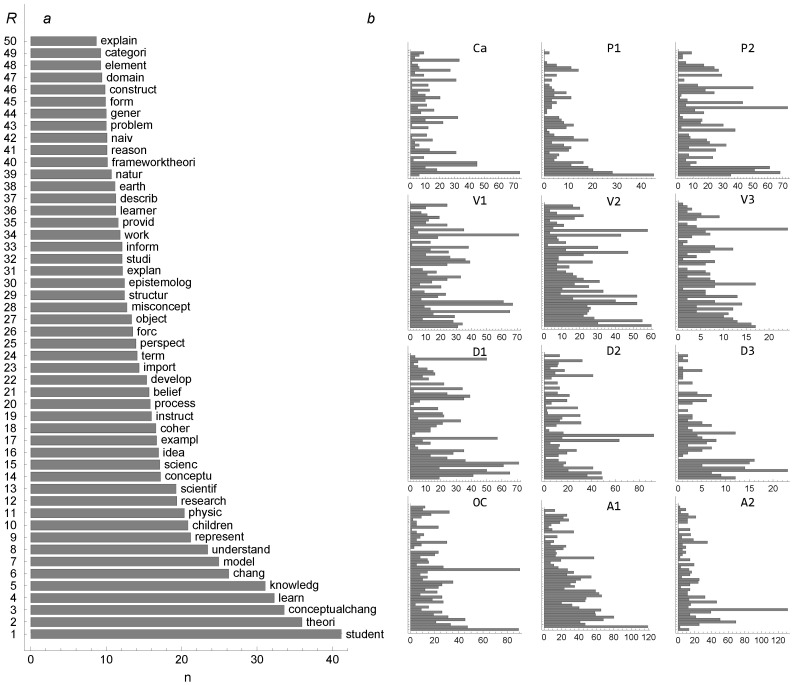
The distributions of occurrence (number) frequency *n* of terms. The rank *R* according to number frequency *n* is shown in panel (**a**) on the left for the 50 most frequently appearing key terms in all 12 texts. The distributions of frequencies in individual texts (denoted by acronyms referring to Table 1) are shown on the right in panel (**b**), in the same order as in (**a**).

**Figure 2 entropy-24-01058-f002:**
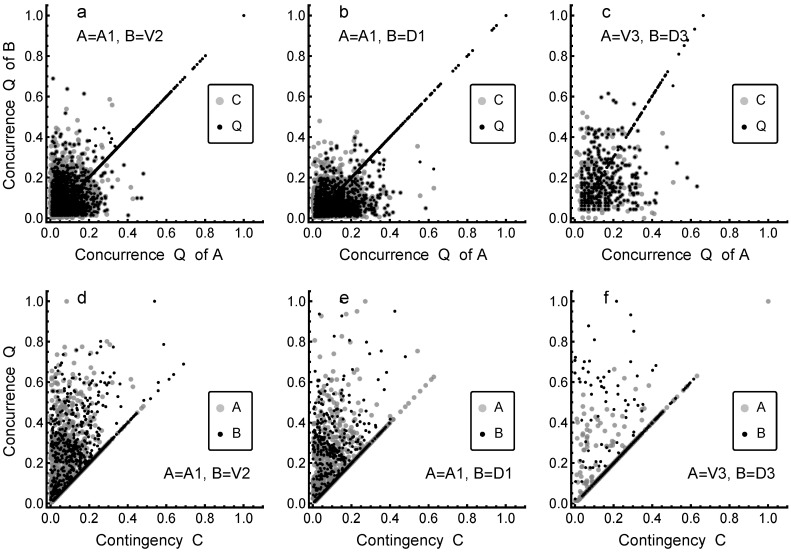
Optimized concurrences’ (*Q*) corresponding contingencies *C* compared. (**a**–**c**) (upper row) display three examples in the form of a scatter plot showing the optimized concurrences *Q* (black symbols) and corresponding contingencies *C* (gray symbols, in the same scale, although the axis label is only for *Q*) for lexicons A and B. The three cases are indicated by acronyms referring to texts (see Table 1). (**d**–**f**) (lower row) display the same cases as (**a**–**c**) but plot the values of concurrence *Q* against values of contingency *C* for each lexicon.

**Figure 3 entropy-24-01058-f003:**
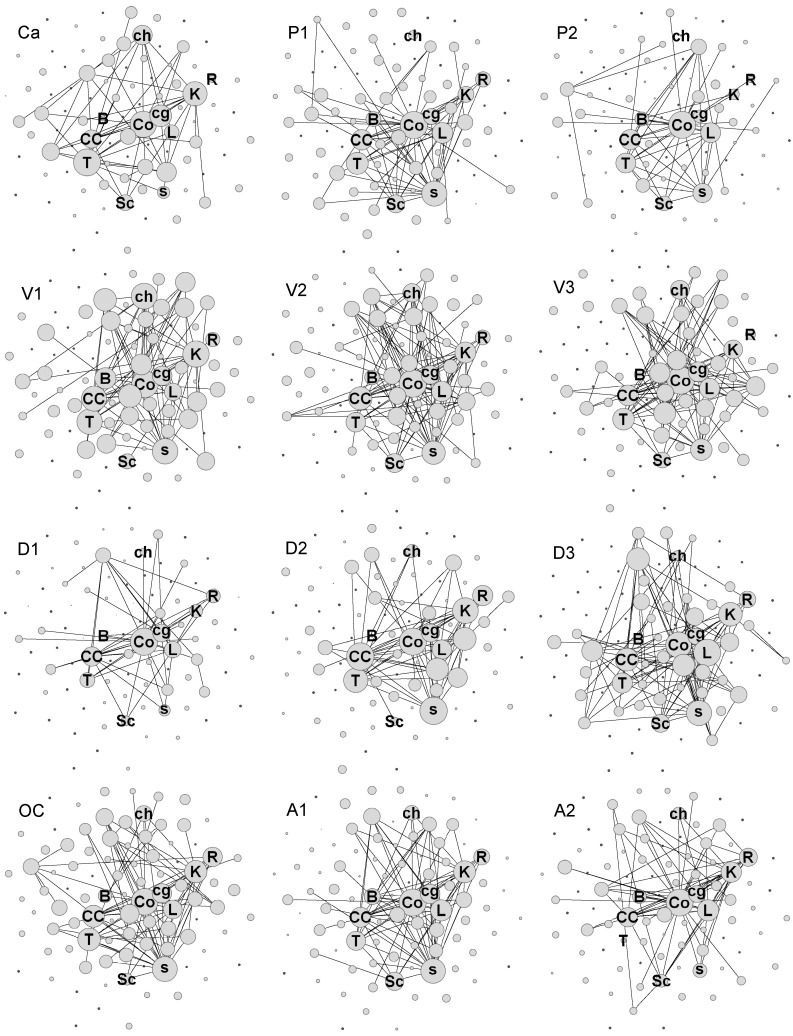
The lexicons of 12 research articles on conceptual change. The 12 lexicons visualized as spring-embedded networks. The acronyms of lexicons (upper left corner) identify the texts, as summarized in Table 1. Figure provides a schematic view, and therefore, only a few nodes are identified: Co = concept, CC = conceptual change, cg = change, ch = child, B = belief, K = knowledge, L = learn, R = research, s = student, and Sc = science. In each lexicon, a given node is always in the same position. The size of each node is proportional to its correlation centrality (as defined in Equation (Equation 5)).

**Figure 4 entropy-24-01058-f004:**
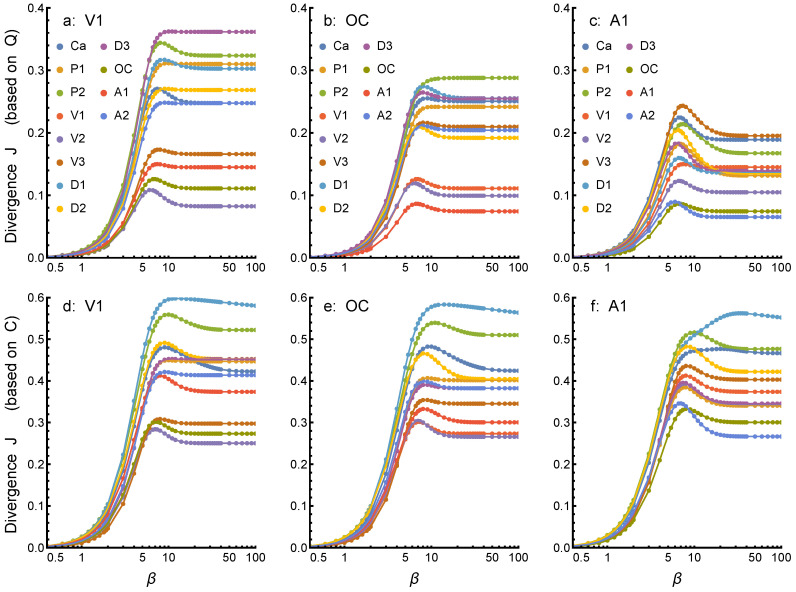
Jensen–Shannon divergences *J* for three lexicons, V1, OC, and A1. The divergence J(A||B) with increasing β is shown in panels (**a**–**c**) (upper row) for lexicons V1, OC, and A1, based on concurrence *Q*; and in panels (**d**–**f**) (lower row) based on contingency *C*. In each case, divergence *J* for a given lexicon A as compared to all other 11 lexicons B is as indicated by acronyms in legends.

**Figure 5 entropy-24-01058-f005:**
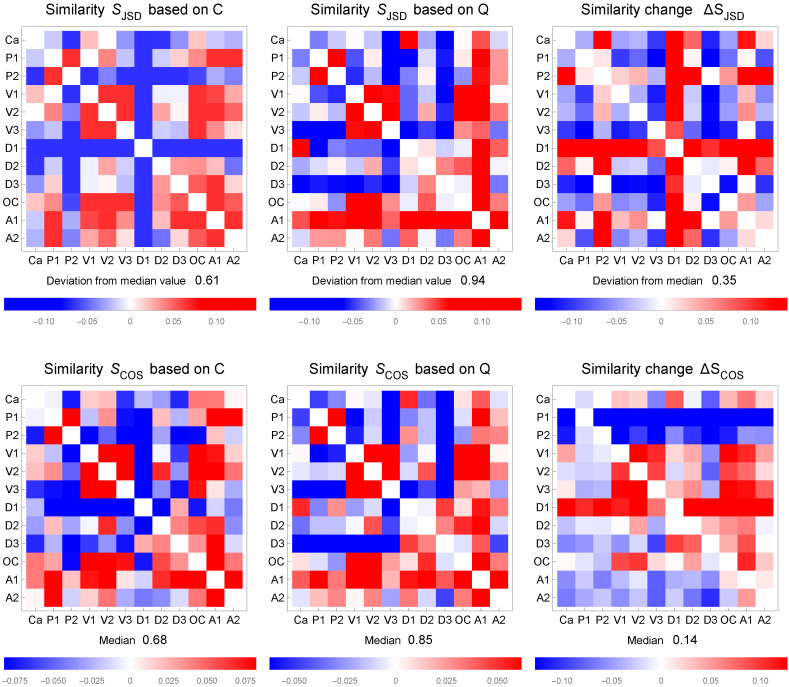
Similarity comparisons of lexicons. The similarity $S_{\mathrm{JSD}}$ based on Jensen–Shannon divergence is shown in the upper row for contingency *C* (at left) and concurrence *Q* (middle). The changes ΔS due to tuning are also shown (right). All values are shown as deviations from the median (value provided above bar-legend). The cosine similarity $S_{\mathrm{COS}}$ is shown in the lower row. In both cases, results are for a stabilized state with β=50.

**Figure 6 entropy-24-01058-f006:**
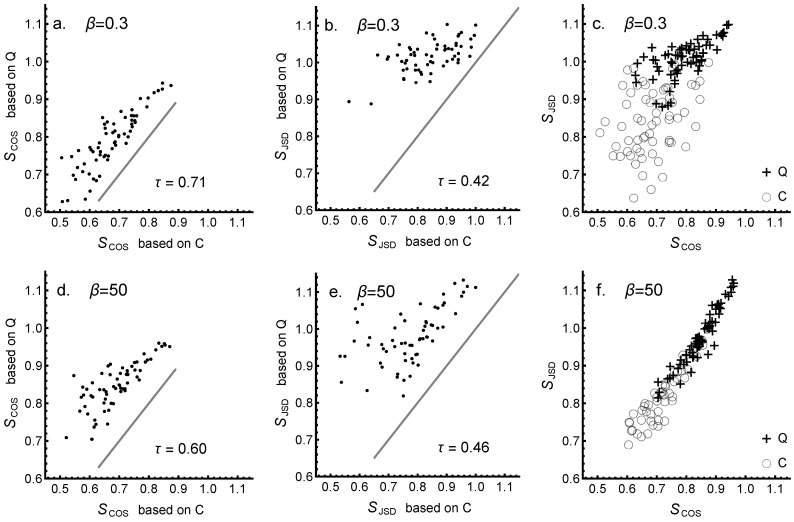
Scatter plots of similarities $S_{\mathrm{JSD}}$ and $S_{\mathrm{COS}}$ for all 12 lexicons. Similarities based on concurrence *Q* and contingency *C* ($S_{\mathrm{COS}}$ (left) and $S_{\mathrm{JSD}}$ (in the middle). Straight lines indicate the references of equal value. Values of (non-parametric) Kendall’s tau correlation coefficient τ are provided. The values corresponding to the initial state β=0.3 are shown in the upper row, and those corresponding to the final stabilized state β=50 in the lower row. The right panel shows scatter plots of $S_{\mathrm{JSD}}$ against $S_{\mathrm{COS}}$.

**Figure 7 entropy-24-01058-f007:**
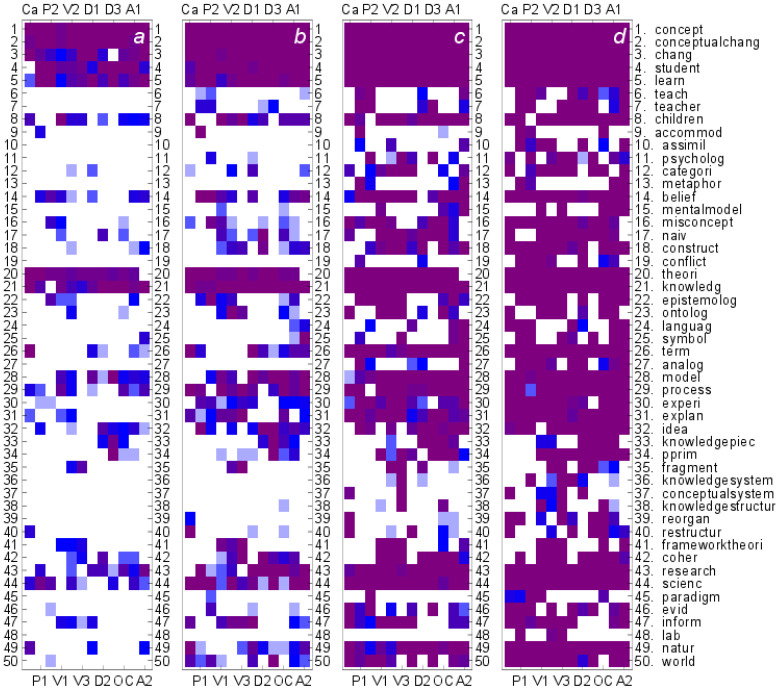
Patterns of shared terms in optimized lexicons based on concurrence *Q*. Occurrences of terms in top 10 (**a**), 20 (**b**) 40 (**c**), and 60 (**d**) cohorts in all 12 cases in the stabilized case with β=50. When a term in the lexicon occurs in all 11 tuned cases, the symbol is purple; otherwise blue. The lighter the symbol, indicating occurrence, the fewer the tuned lexicons where the term occurs. Note that only the terms occurring at least eight times are shown.

**Figure 8 entropy-24-01058-f008:**
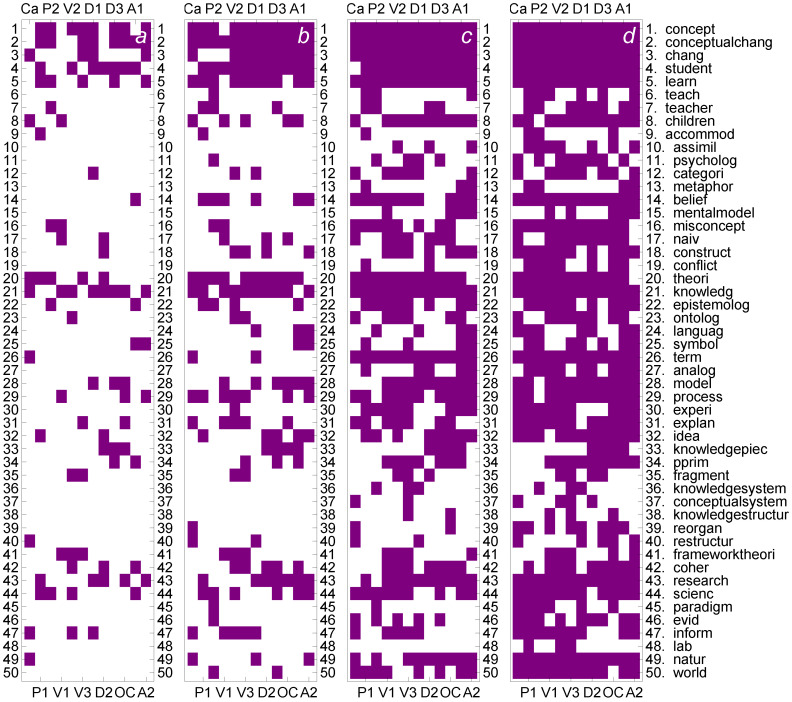
Patterns of shared terms in lexicons based on concurrence *C*. Occurrences of terms in top 10 (**a**), 20 (**b**) 40 (**c**), and 60 (**d**) cohorts in all 12 cases in the stabilized case with β=50. Terms occurring at least eight times are shown.

**Table 1 entropy-24-01058-t001:** Scholarly text about conceptual change used as a sample. Acronyms (Acr.) specify the text provided as a reference, which is briefly described in last column.

Acr.	Ref.	Description of Position
Ca	[55]	Developmental psychology and cognition; theories of concepts; cognitive science
P1	[56]	Piaget’s theory of learning; Kuhn’s theory of science; scientific knowledge
P2	[57]	Updated and augmented version of P1
V1	[58]	Framework theory; Knowledge-as-theory view; synthetic conceptual models
V2	[59]	Updated and extended version on V1; Contrasts to Knowledge-as-elements view
V3	[60]	Review of views in V1 and V2; Contrasts to Knowledge-as-elements view
D1	[61]	P-prims and knowledge-as-elements view; Contrasts to knowledge-as-theory view
D2	[62]	Augments D1 by Coordination classes; Contrasts to knowledge-as-theory view
D3	[63]	Review of views D1 and D2; Contrasts to knowledge-as-theory view
OC	[64]	Attempts synthesis between knowledge-as-elements and -theory views
A1	[65]	Discusses views Ca, P1-2, V1-2, D1-2 and attempts to find common viewpoint
A2	[66]	Review, augments A1, with specific emphasis on views as in Ca

**Table 2 entropy-24-01058-t002:** Summary of symbols and abbreviations used recurrently in the text and figures.

Symbol/Abbreviation		Symbol/Abbreviation		Symbol/Abbreviation	
*Q*	Concurrence	Q	Concurrence matrix	β	Time-like parameter
$Q_{0}$	Prefactor of *Q*	${\left[Q\right]}_{ij}$	Element ij of Q	Γ	Correlation matrix
*R*	Ratio of averages in *Q*	C	Contingency matrix	$\Gamma_{i}$	Correlation centrality
Θ	Phase factor in *Q*	${\left[C\right]}_{ij}$	Element ij of C	$\bar{\Gamma }$	Vector of elements $\Gamma_{i}$
$n_{xy}$	Co-occurrence frequency	W	Normalized W	*H*	von Neumann entropy
W	Weighted adjac. matrix	${\left[W\right]}_{ij}$	Element ij of W	*J*	J-S divergence (JSD)
${\left[W\right]}_{ij}$	Element ij of W	ρ	Density matrix	$S_{\mathrm{JSD}}$	JSD-similarity
$d_{i}$	Weighted node degree	${\left[\rho \right]}_{ij}$	Element ij of ρ	$S_{\mathrm{COS}}$	cos-similarity

**Table 3 entropy-24-01058-t003:** Numbers of sentences ($N_{S}$) and words ($N_{W}$) and their ratio $N_{W}/N_{S}$ in texts listed in Table 1.

	Ca	P1	P2	V1	V2	V3	D1	D2	D3	OC	A1	A2
$N_{S}$	339	160	453	240	245	86	626	392	105	260	410	287
$N_{W}$	3801	1617	3793	3299	3449	1205	5692	3720	1075	3092	6185	3651
$N_{W}/N_{S}$	11.2	10.1	8.4	13.7	14.1	14.0	9.1	9.5	10.2	11.9	15.1	12.7

**Table 4 entropy-24-01058-t004:** Cumulative appearance of terms in cohorts of top 10 (I), 20 (II), 40 (III), and 60 (IV) terms, for concurrence (Q) and contingency (C)-based ranking. Terms that appear in at least 8 lexicons are listed. In the column for β=0.3, terms that also appear for β=50.0 are in boldface, and in the corresponding column, only additional terms are listed.

Cohort	β=0.3	β=50
Q-I	**concept**, **conceptual change**, **knowledge**, **student**	theory/ies
Q-II	**change**, **learn**	process
Q-III	**belief**, **children**, **explanation**, **model**,	
	**research**, **science**, **term**, process	nature, idea
Q-IV	**construct**, **epistemol.**, **experience**	naive
	**information**, **misconception**, **p-prim**, **world**	
	nature, idea, analog, coherence	
C-I	**concept**, student	
C-II	**change**, **learn**,**conceptual change**, **knowledge**, **thery/(ies)**	student
C-III	**belief**, **children**,**research**, **science**, **term**,	
	**process**,**nature**, explanation, idea	
C-IV	**construct**, **epistemol.**, **experience**	naive, idea
	**information**, **misconception**, **p-prim**, **world**	explanation
	**analog**, **coherence**, **evidence**, model	

## Data Availability

Not applicable.

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
