# Peer review of "Lexicons of Key Terms in Scholarly Texts and Their Disciplinary Differences: From Quantum Semantics Construction to Relative-Entropy-Based Comparisons"

_entropy, 2022, doi:10.3390/e24081058_

Round 1
Reviewer 1 Report
The method is innovative and the experiment is very interesting to demonstrate the potential of quantum semantics from a philosophical and epistemological point of view. It would be interesting perhaps to add some lines to provide an interpretation of quantumness, concurrence, and JSD in terms of linguistic notions.
Author Response
We thank reviewers for their valuable comments and suggestions. We have tried to respond as well as we are able to all comments and suggestions by reviewer 2. Reviewer 1 had no specific comments, but we believe that our responses to comments by reviewer 2 also cover the general suggestions by reviewer 1. In two cases, we found no appropriate way to revise the manuscript and we have provided reasons for that. We hope our arguments in these cases are found appropriate. Detailed responses are in the attached pdf-file.

Reviewer 2 Report
The authors produced an innovative investigation of texts through novel techniques grounded in co-occurrence networks and quantum physics. They provide novel analytical implementations to quantum entanglement as a way to capture how individuals reconstruct meaning from word associations in texts.
The manuscript is well written and contains interesting results about texts revolving around conceptual change. I have a few comments that could improve manuscript quality and recommend this paper for publication in Entropy after minor revisions.
The authors should be praised for bridging concepts across communities in a truly transdisciplinary way.
---
Introduction - Should quantum semantics be discussed in an appropriate subsection?
Line 45 - Why word co-occurrences and not syntactic relationships? Teixeira and colleagues (https://doi.org/10.1038/s41598-021-98147-w) show how co-occurrences and syntactic parsing with NLP can give rise to networks highlighting both similar and distinctive patterns. Could the methodology presented by the authors on co-occurrence networks be extended also to the methods reviewed in 63-78 and in Teixeira et al (2021)? This point might be used to better close that paragraph or to enrich the discussion for future research, since co-occurrences miss many syntactic relationships giving meaning to words in sentences.
Line 49 - Thus, such lexicon thus become -> Such lexicon thus becomes...
Line 123 - The density matrix could use a reference.
Lines 148-150 - A bullet point list could help the reader here.
Table 1 - This should be enriched with some text statistics - How many sentences, words, unique words were featured in those texts?
Eq. 1 - A parenthesis is missing. Could the formula be refactored and shortened by adding a second shorter version with Q, R? This would improve clarity.
268 - lings -> links
291 - Missing reference.
330-350 - This part should be moved to the Methods. Mathematica's implementation of stemming uses Porter's algorithm, so a reference to that method should be added.
365 - Explaining a bit more why the optimised concurrence is obtained would help the reader interpreting Figure 2 more easily. That part should also be split in another subsection separate from the frequency analysis.
Figure 3 - In its current state, this figure is a bit problematic. We can visually see different topologies but without visualising nodes' names, not much info is acquired here. What about using the frequency analysis to select say top X (e.g. X = 10) most central concepts and plot them to label nodes in Figure 3? In that way, authors would still avoid issues with labelling too many nodes while providing some more information.
Figure 4 - The peaks in Figure 4 are quite intriguing. What are they relative to? Are they an effect of network size or exploration under the density matrix formalism of random explorations of network topology? Would it be significant to observe Figure 5-like results when \beta is selected to fit those peaks?
Line 443 - This aspect should be underlined more. Would the same conclusion emerge by having a look only at the frequency distributions?
Figure 7 - How is the colour scheme defined here?
Line 545 - According to the results in Figures 7 and 8, can "tunable" mean how "generic" or "well-explained" are those terms in the considered texts? Could Q and C be used to capture word-level aspects like "familiarity" or "concreteness" (see Scott et al. 2019, https://doi.org/10.3758/s13428-018-1099-3). If a text defines a concept in a "complete" way, leading to no ambiguity across readers, could this aspect be captured by Q and C? More concrete or familiar terms, if explained together with other concrete and familiar terms, might correspond to such "completely"-defined terminologies. I do not expect a quantitative analysis of this but wonder whether the results obtained by authors could be framed in this intriguing way.
Line 565 - Reference missing to quantum semantics.
Conclusions - Some limitations of this work have been acknowledged but what about emotional profiles? Could quantum semantics be susceptible to the emotional ways in which the same concepts can be described differently and also interpreted differently by readers? The work by Surov (2021) 10.20944/preprints202111.0379.v1 could serve to write 1-2 sentences about this limitation.
Last but not least, the manuscript contains a few typos, I suggest a careful check of the text.
Author Response

(The authors gave the same response as above.)
